# Awareness of Medical Doctors in Pusat Perubatan Universiti Kebangsaan Malaysia on Diagnostic Radiological Examination Related Radiation Exposure in the Pediatric Population

**DOI:** 10.3390/ijerph19106260

**Published:** 2022-05-21

**Authors:** Chee Guan Ng, Hanani Abdul Manan, Faizah Mohd Zaki, Rozman Zakaria

**Affiliations:** 1Department of Radiology, Faculty of Medicine, Hospital Canselor Tuanku Muhriz, Universiti Kebangsaan Malaysia Medical Centre, Kuala Lumpur 56000, Malaysia; henryncg@gmail.com (C.G.N.); drfaizah@ppukm.ukm.edu.my (F.M.Z.); rozman@ppukm.ukm.edu.my (R.Z.); 2Hospital Wanita Dan Kanak-Kanak Kuala Lumpur, Hospital Kuala Lumpur, Jalan Pahang, Kuala Lumpur 50586, Malaysia; 3Functional Image Processing Laboratory, Department of Radiology, University Kebangsaan Malaysia Medical Centre, Cheras, Kuala Lumpur 56000, Malaysia; 4Department of Radiology and Intervency, Hospital Pakar Kanak-Kanak (Specialist Children Hospital), University Kebangsaan Malaysia, Cheras, Kuala Lumpur 56000, Malaysia

**Keywords:** pediatric, radiation protection, healthcare professional

## Abstract

Background: When exposed to equal radiation doses, the risks for children and adolescents are more significant than for adults. Children grow quickly, and their cells are more sensitive to radiation. After radiation exposure, children have a higher risk of developing malignancies such as leukemia, thyroid abnormalities, and various types of cancers. The healthcare professionals’ (in this context referring to medical doctors at all levels) awareness of imaging modalities associated with ionizing radiation is essential to ensure optimal patient management of cooperation in dealing with radiation exposure. Therefore, the present study is aimed to evaluate the awareness of healthcare professionals on medical imaging-related radiation exposure in the pediatric population in our center, Pusat Perubatan Universiti Kebangsaan Malaysia. Materials and Methods: A cross-sectional survey was conducted among healthcare professionals using self-administered validated questionnaires in a university hospital for a duration of seven months. Healthcare professionals of all levels participated in this survey. Results: A total of 145 healthcare professionals participated in this study. More than half of the respondents are house officers, 57.2% (*n* = 83). Results indicated that only 6 out of 145 healthcare professionals who participated in this survey had attended a radiation protection course. This survey showed that 37.2% of the respondents were unaware that chest radiographs would expose patients to ionizing radiation. Finally, results also indicated that senior doctors (21 out of 24 participants) showed better awareness of radiation protection knowledge. Conclusions: In general, healthcare professionals in our institution are inadequate in awareness of medical radiation exposure, particularly among house officers. However, the awareness of radiation safety and exposure improves with the number of years of clinical practice. We propose that some younger healthcare professionals do not take radiation safety seriously. Moreover, we would like to suggest all healthcare professionals must attend a radiation safety course, as we expect this will improve patient outcomes.

## 1. Introduction

Hiroshima and Nagasaki are the most significant nuclear disasters in human history, providing clear evidence that ionizing radiation is a human carcinogen. Data show significant increases in the blood, breast and other cancers have been observed in the bomb survivors [1]. These epidemiological studies have also demonstrated that exposure to ionizing radiation during childhood may increase cancer risk compared with adults, pointing toward higher sensitivity to radiation-induced cancers in the pediatric population. Among exposed children, the incidence of leukemia rose dramatically just a few years after exposure [2]. Studies from the Chornobyl incident further expanded our knowledge on the increased sensitivity of children to ionizing radiation, as indicated by the substantial increase in the relative risk of thyroid cancer in children exposed to high thyroid doses (over 1 Gy) in their childhood [3,4]. Therefore, it is now generally accepted that children are more sensitive to radiation than adults, specifically with a higher relative risk of cancers, including leukemia, brain, breast, skin, and thyroid cancers following exposure [3,5]. This is partly because of the radiosensitivity of their developing organs and tissues [6,7]. Additionally, the longer post-exposure life expectancy increases the lifetime risks of developing radiation-induced malignancies [8,9,10]. This is becoming important because 70% to 80% of all children diagnosed with cancer have long-term survival [5,11,12].

Technology in medical imaging has increased dramatically in the recent decade. With the growth in availability, diagnostic radiological imaging requests have also increased in recent years [13,14]. From 1997 to 2007, more than 3.6 billion diagnostic medical imaging were performed annually worldwide [13,15]. Approximately 3–10% of all radiological procedures were performed in children with variable frequency among countries [13,16]. Some medical imaging is associated with ionizing radiation, such as general radiography, computed tomography (CT) and fluoroscopy [16]. Based on a report from the American Cancer Society [12], the US Food and Drug Administration (FDA) estimates that exposure to 10 mSv from an imaging test would be expected to increase the risk of death from cancer by about 1 chance in 2000 [17,18]. Knowing this, the application of the radiation protection principles proposed by the International Commission on Radiological Protection (ICRP) has become more critical, particularly for the pediatric population [19,20]. In this context, the radiation protection principles refer to justification, optimization, and a system of dose limits [5,21]. Radiological medical practitioners (RMP) [13], such as radiologists, often enforce radiation protection principles. However, the referrer’s role in upholding these principles is equally important. “Referrer” refers to a healthcare professional who requests diagnostic medical imaging [14], such as a pediatrician or pediatric surgeon. The referrer could reduce the radiation dose received by patients through discussion with RMP on the most suitable and the safest radiological examination to be performed [18]. In the whole manuscript, healthcare professionals in this context refer to medical doctors, including consultants, specialists, medical officers, and house officers. The present study’s house officer refers to a medical practitioner undergoing internship training under the Medical Act 1971.

A previous study by Quinn et al., 1997 shows that despite the POPUMET regulations, most clinicians have not received adequate radiation protection teaching. Even if a course has been attended, overall knowledge is still poor [22]. Another study also shows the same results, suggesting that the level of radiation protection awareness among health care professionals is insufficient to ensure patient and worker safety [23]. With increased public awareness of medical radiation exposure, communication between the healthcare professional and patient (in this context, refers to parent or caretaker) is essential [19]. Healthcare professionals should be knowledgeable and be able to explain to the patients about radiation risk.

Furthermore, patients have the right to know the radiation dose they receive for every imaging procedure [20]. F. Ria et al. reported that 56.4% of the patient respondents in the study did not know about ionizing radiation-related imaging modalities [9]. Moreover, to protect patients from an excessive radiation dose, there is a paradigm shift from the concept of radiation protection through nationwide collection and reduction in radiation dose to individual dose and exposure tracking [24]. In 2006, the International Atomic Energy Agency (IAEA) introduced the Smart Card project to track cumulative radiation doses [25]. This program facilitates both clinicians and radiologists to decide on suitable imaging modalities for patients [26,27,28,29].

All departments can request these diagnostic examinations in our institution, and all are submitted and endorsed by the Radiology Department. Therefore, it is crucial to evaluate the level of awareness among healthcare professionals on medical radiation exposure used in the pediatric population in our institution. In addition, we also include a survey to determine the respondents’ opinion on the initiation of a cumulative radiation dose-tracking program for pediatric patients.

## 2. Materials and Methods

A cross-sectional survey was conducted among healthcare professionals using self-administered questionnaires in a university hospital for seven months, from June 2020 to December 2020. The study protocol (FF-2018-439) was approved by an institutional, local ethics committee. The main objective of this study is to determine the level of knowledge among healthcare professionals on medical radiation exposure used in the pediatric population.

### 2.1. Study Population

A total of 145 healthcare professionals of all levels, including consultants, specialists, medical officers and house officers involved in the management (either directly, indirectly or with prior experience) of pediatric patients, were included in this study. We had sent 250 invitations to health professionals to respond to our study in our institution, Pusat Perubatan Universiti Kebangsaan Malaysia.

In particular, house officers were invited to participate, as pediatric attachment is compulsory for internship training in this country (Malaysia) with a minimum of 4 months. To achieve a confidence level of 95% with a tolerated margin of error within 5%, the total number of respondents needed was 108 doctors. Any healthcare professionals who refused to give consent or sent incomplete questions were excluded from this study.

### 2.2. Questionnaire

The questionnaire was designed with joint input from all authors, which included two pediatric radiologists, a physicist, and a biostatistician. The question was based on a previous similar study by Kew et al., 2012 [27]. The questionnaire was piloted using other medical professionals, such as radiographers, whose work involved radiation. It was composed of questions that required a simple tick and short answer. The knowledge-based questions were deemed statistically reliable and valid for the objectives. The internal consistency test’s reliability analysis was acceptable with a Cronbach’s alpha value of 0.775 (95% CI: 0.727–0.818). The validity testing was analyzed using the item difficulty index, item discrimination index, and exploratory factor analysis. It showed that the knowledge-based questions had different difficulty levels with acceptable discriminating characteristics.

The questionnaire was divided into two sections. In Section A, respondents were asked questions regarding demographic information. For example, gender, years of experience in medical service, specialty, and respondents’ experience attending a radiation protection course. In Section B, there were 17 knowledge-based questions. The first question was on the perception of cancer risk in children exposed to a single abdominal CT scan. Subsequently, there were 11 questions related to the identification of medical imaging associated with ionizing radiation. The last 5 questions were related to the estimation of effective doses in terms of a chest radiograph equivalent for commonly used radiological imaging in the pediatric population. There were questions to determine the presence of communication between pediatric healthcare professionals, parents and caretakers on ionization radiation risk as well as to ascertain the respondents’ agreement on the radiation-tracking program.

### 2.3. Data Collection

This study was carried out from 1 June 2020 to 31 December 2020. Self-administered questionnaires were distributed to the healthcare professionals via a Google sheet form or hardcopy during this period. The respondents were requested to answer all the questions in one setting without any references.

### 2.4. Analysis and Statistical Method

Statistical analysis was performed using IBM SPSS Version 22. Descriptive analyses such as frequency and percentage were used to determine the level of knowledge on ionizing radiation-related imaging modalities, effective dose estimation, and cancer risk perception. To evaluate the overall level of knowledge, an arbitrary one point was given to each correctly answered knowledge-based question. No points were awarded for an incorrect answer. Then, further analyses were performed using chi-square and Spearman’s correlation. These analyses were used to determine whether there was an association between knowledge level and different variables such as gender and respondents’ years of service in the medical field. The statistical significance level was set as a *p*-value < 0.05.

## 3. Results

### 3.1. Respondents’ Socio-Demographic Characteristics and Prior Experience on Radiation Protection Concept

A total of 145 healthcare professionals participated in this study (*n* = 145), with 100 female participants (69%) and 45 male participants (31%). House officers made up 57.2% of the total respondents (*n* = 83). The remaining respondents included 38 medical officers and 24 specialists. More than 25% of the participants (*n* = 41) were from pediatric and pediatric surgery specialties (Table 1). Results also indicated that 27 respondents (18.6%) were reported to have heard of radiation protection principles, for example, as low as reasonably achievable (ALARA). However, only 6 participants (4.1%) out of the 145 healthcare professionals had formally attended a radiation protection course.

### 3.2. Perception of Cancer Risk in the Pediatric Population and Awareness of Medical Imaging Related Radiation Exposure

On the perception of cancer risk in the pediatric population, only 33.8% of the total respondents (*n* = 49) agreed that the lifetime risk of cancer is higher than in adults as a result of exposure to a single abdominal CT. In the present study, medical imaging is divided into non-ionizing and ionizing radiation-related medical imaging. Non-ionizing medical imaging refers to imaging not associated with ionizing radiation, namely ultrasound and magnetic resonance imaging (MRI). The chest radiograph is a commonly requested ionizing radiation-related medical scan in our center. More than 35% of the total respondents were unaware that performing chest radiographs would expose pediatric patients to ionizing radiation (Table 2). Results also show that more than 70% of the respondents correctly answered that an abdomen ultrasound, a spine ultrasound, and a lower limb Doppler ultrasound used a non-ionizing imaging modality. However, some respondents failed to recognize that these examinations belong to the same modality used to investigate different body parts (Table 3). Most of the respondents (more than 50%) answered no to the questions they did not know or were unsure about (wrong answer).

### 3.3. Effective Dose Estimation

Less than a quarter of the respondents could correctly estimate the effective dose for some commonly requested imaging in an average 5-year-old child. This imaging included CT head, CT chest, CT abdomen, FDG PET CT, Tc-99m bone scan and fluoroscopic cystogram (Table 4).

### 3.4. Overall Knowledge of Medical Imaging Radiation Exposure in the Pediatric Population

The median score for the 17-item knowledge-based questions was 9.0. Authors arbitrarily set the satisfactory knowledge level as respondents who could answer more than eight knowledge-based questions (out of 17). Only 78 participants (53.8%) achieved satisfactory knowledge levels, with a significant association demonstrated between the respondents’ seniority and their level of knowledge (Spearman Correlation = 0.378, *p*-value = 0.000). In general, specialists and medical officers had better awareness than house officers on medical imaging-related radiation exposure for the pediatric population (Table 5). Similarly, healthcare professionals with more than ten years of experience showed better awareness of medical imaging-related radiation exposure. (Spearman Correlation = 0.379, *p*-value = 0.000) (Table 6). This present study also showed that pediatric and pediatric surgery specialty respondents had better knowledge of this subject matter than other specialties (Pearson’s chi-square = 19.520, *p*-value = 0.000) (Table 7). No association was demonstrated between the level of knowledge or gender in this study.

### 3.5. Communicating Radiation Risk

Lack of communication between healthcare professionals and patients regarding ionizing radiation dose exposure and the associated risk was highlighted by A. Ribeiro et al. [16]. In our study, it was reported that only 63.4% of the healthcare professional respondents explained the associated risks of radiological examinations to parents and caretakers. Generally, healthcare professionals who have better knowledge of this subject matter tend to explain to parents and caretakers, as they are more likely to realize the importance of risk communication (Spearman’s correlation = 0.193, *p*-value = 0.020), refer Table 8.

## 4. Discussion

### 4.1. Awareness of Non-Ionizing and Ionizing Radiation Related Imaging Modalities

Plain radiography, such as chest radiograph, is frequently requested for the pediatric population in our center. Even though the individual’s radiation risk is low for plain radiography, considerable collective risk cannot be ignored [30]. Surprisingly, more than 35% of the respondents were unaware that chest radiography would expose pediatric patients to ionizing radiation.

Non-ionizing modalities such as ultrasonography and MRI are preferred in the pediatric population whenever indicated and available [13]. Approximately 75% of the respondents were aware that an abdominal ultrasound, a lower limb Doppler ultrasound, and spinal ultrasound use a non-ionizing imaging modality. However, this result was less than the result found in a study by T. Y. Kew et al. [28], which revealed 96% of their respondents were able to identify ultrasound as a non-ionizing procedure. There are also respondents (*n* = 10) that failed to recognize an abdominal ultrasound, a lower limb Doppler ultrasound, and spinal ultrasound as having the same physical properties. It is alarming that only 49% of our respondents realized that MRI is non-ionizing compared with 66% in the study mentioned above. A postulated explanation for this is that our current undergraduate medical education focuses more on disease recognition with a lack of emphasis on radiation protection elements, as indicated in a previous study from other institutions [18].

### 4.2. Dose Estimation and Awareness of Cancer Risk

To communicate with the public about radiation used in medical imaging, a comparison with the effective dose concept might be useful [20]. For example, expressing the effective dose in multiple commonly known examinations such as the chest radiograph [20]. Less than 15% of the respondents in this study could correctly estimate the effective dose of commonly requested imaging procedures for pediatric patients. In addition, only 41.9% of the respondents realized that ionizing radiation would increase the lifetime risk of developing cancer among pediatric patients. One possible explanation is that our respondents had no prior exposure to medical radiation information and were unaware of radiation protection principles. This is evidenced as more than 95% of the respondents had never attended a formal radiation protection course. About 80% of them were unaware of the ALARA concept. This is similar to the study by Al-Rammah [2] that reported 85% of their pediatrician respondents were also unfamiliar with the ALARA concept.

### 4.3. Overall Knowledge of Medical Imaging Radiation Exposure in the Pediatric Population

It was reported that only 53.8% of the total respondents had satisfactory knowledge levels. It is worth noting that junior doctors, particularly house officers, showed low awareness of this subject matter. This needs to be addressed as they are involved in explaining medical imaging to the parents and caretakers.

There is a significant association between respondents’ professional grade as well as years in service and their level of knowledge. Hence, it is instead assuring that specialists and medical officers showed better awareness of the subject matter than house officers. One possible reason postulated was that experience gained over the years during medical practice would improve knowledge level in addition to more communication with RMP that could have transferred the radiation knowledge when discussing the best imaging modality for their patients’ management.

We recommend that healthcare professionals, especially junior doctors, be educated on medical imaging-related radiation exposure and radiation protection principles. This could be implemented through participation in a radiation protection course organized by the radiology department. In addition, to ensure senior doctors, particularly specialists, are more aware of their role as referrers, radiation protection principles could be incorporated into the undergraduate and postgraduate medical syllabi [28]. Most countries make it compulsory for healthcare professionals to attend a radiation protection course. Some countries also expect their healthcare professionals to pass the radiation protection examination.

### 4.4. Acceptance of Medical Radiation Tracking Program among Healthcare Professionals

Approximately 90% of the health professionals in this study support a medical radiation-tracking program, which helps them decide on a more suitable radiological examination, trace old examination records, as well as prevent unnecessary and repetitive examinations. For effective radiation dose tracking, the ultimate way is to implement a radiation-tracking program. This study has shown that our respondents were supportive of this program. Thus, a further plan should be implemented to have this program in place in the future.

### 4.5. Limitation

The present study used convenience sampling, which may result in sampling bias. The larger sample size from senior doctors, such as medical officers and specialists, would better represent the overall knowledge level. In addition, this study was conducted in one center (in a single university hospital); hence, the result may not be representative of the whole country. The results could not be transposed to other hospitals in different countries. However, results are at least significant for the hospital we are working in and may help raise awareness of radiation protection and support the importance of education on radiation protection. A multi-center survey is recommended for future studies to generate a more representative result.

## 5. Conclusions

Only half of the respondents achieved a satisfactory knowledge level, with a significant association between the respondents’ seniority and their level of knowledge. Therefore, there is an urgent need for a change in the education of medical students and young doctors to improve awareness of radiation exposure in diagnostic radiological examinations. Furthermore, incorporating radiation protection principles in the undergraduate medical student curriculum is advisable to equip our future doctors with this knowledge.

## Figures and Tables

**Table 1 ijerph-19-06260-t001:** Distribution of respondents’ specialty/current posting.

Specialties/Current Posting	Number of Participants	Percentage (%)
Pediatric	31	21.4
Pediatric Surgery	10	6.9
Medical	28	19.3
General Surgery	18	12.4
Obstetrics and Gynecology	6	4.1
Orthopedic	14	9.7
ENT	7	4.8
Ophthalmology	1	0.7
Oncology	1	0.7
Family Medicine	3	2.1
Other	26	17.9
Total	145	100.0

**Table 2 ijerph-19-06260-t002:** Summary of questionnaire responses on ionizing radiation-related medical imaging.

Questions	Number of Respondents
1. Chest Radiograph Uses Ionising Radiation?	
Yes	91 (62.8%)
No/Unsure	54 (37.2%)
2. Upper Gastrointestinal Fluoroscopic Study Uses Ionising Radiation?	
Yes	86 (59.3%)
No/Unsure	59 (40.7%)
3. CT Thorax Uses Ionising Radiation?	
Yes	115 (79.3%)
No/Unsure	30 (20.7%)
4. Tc-99m Meckel Scan Uses Ionising Radiation?	
Yes	87 (60.0%)
No/Unsure	58 (40.0%)
5. HRCT Temporal Bone Uses Ionising Radiation?	
Yes	101 (69.7%)
No/Unsure	44 (30.3%)
6. Diagnostic Cerebral Angiogram Uses Ionising Radiation?	
Yes	84 (57.9%)
No/Unsure	61 (42.1%)

**Table 3 ijerph-19-06260-t003:** Summary of questionnaire responses on dose-saving medical imaging.

Questions	Number of Respondents
1. Ultrasound Abdomen Uses Ionising Radiation?	
No	110 (75.9%)
Yes/Unsure	35 (24.1%)
2. MRI Brain Uses Ionising Radiation?	
No	74 (51.0%)
Yes/Unsure	71 (49.0%)
3. Ultrasound Spine Uses Ionising Radiation?	
No	110 (75.9%)
Yes/Unsure	35 (24.1%)
4. Ultrasound Doppler Lower Limb Uses Ionising Radiation?	
No	108 (74.5%)
Yes/Unsure	37 (25.5%)

**Table 4 ijerph-19-06260-t004:** The number of respondents that correctly estimated the chest radiograph equivalent effective dose for ionizing radiation-related medical imaging in an average 5-year-old child.

Imaging	Number of Respondents	Percent (%)
CT head (100 chest radiographs)	41	28.3
CT chest (150 chest radiographs)	39	26.9
CT abdomen (185 chest radiographs)	25	17.2
FDG PET CT (765 chest radiographs)	25	17.2
Tc-99 bone scan (300 Chest Radiographs)	17	11.7
Fluoroscopic cystogram (16 chest radiographs)	22	15.2

**Table 5 ijerph-19-06260-t005:** Number of respondents who achieved a satisfactory score according to professional’s grade.

Professional’s Grade	Level of Knowledge (Out of 17 Knowledge-Based Questions)		Total
	=/<8	>8	
House officer	51	32	83 (57.93%)
Medical officer	13	25	38 (26.20%)
Specialist/consultant	3	21	24 (16.55%)
Total	67	78	145

**Table 6 ijerph-19-06260-t006:** Number of respondents who achieved satisfactory scores according to years in service.

Years in Service	Level of Knowledge (Out of 17 Knowledge-Based Questions)		Total
	=/<8	>8	
<5 years	51	33	84 (57.93%)
5–10 years	14	23	37 (26.20%)
>10 years	2	22	24 (16.55%)
Total	67	78	145

**Table 7 ijerph-19-06260-t007:** Comparison of knowledge levels between pediatric and pediatric surgery respondents and other specialties.

Specialties	Level of Knowledge (Out of 17 Knowledge-Based Questions)		Total
	=/<8	>8	
Pediatric/pediatric surgery	7	34	41 (28.28%)
Other specialties	60	44	104 (71.72%)
Total	67	78	145

**Table 8 ijerph-19-06260-t008:** Number of respondents who explained to parents the risks associated with radiological examination according to the level of knowledge.

Do You Explain to Parents the Radiological Examination Risk?	Level of Knowledge (Out of 17 Knowledge-Based Questions)		Total
	=/<8	>8	
Don’t know	11	2	13
No	17	23	40
Yes	27	65	24
Total	55	90	145

## Data Availability

Not applicable.

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
