# Peer review of "Awareness of Medical Doctors in Pusat Perubatan Universiti Kebangsaan Malaysia on Diagnostic Radiological Examination Related Radiation Exposure in the Pediatric Population"

_ijerph, 2022, doi:10.3390/ijerph19106260_

Round 1
Reviewer 1 Report
The provided manuscript by Ng et al. deals with the very interesting topic of awareness of health professionals regarding radiation exposure of diagnostic imaging in pediatric patients. Results of this single-center study are at least unexpected, not to say shocking. In the reviewers opinion it is questionable if the results of this study can be transposed to other hospitals in different countries. However, results are at least very important for the hospital the authors are working in and may could help to raise awareness on radiation protection as well as support the importance of education on radiation protection. Before publishing, the manuscript at hand needs several amendments.
Title:
good
Abstract:
p.1 LL30-31 “senior doctors” please provide numbers
p.1 LL32-35 please specify what in your believe are the reasons for the results.
Introduction:
Please be a little bit more specific on the effects of radiation exposure on children; e.g. “damage at the cellular level”? What effect do you expect at what dose? deterministic and stochastic effects? risk assessment of radiation exposure/ linear no-threshold model, what is the cancer risk in the literature? …
Please provide some information on organizational radiation protection measures of your hospital/country; e.g. who can request an examination and who is legally responsible.
Methods:
Please provide the questionnaire.
To how many health professionals did you send the questionnaire? How many do work within your institution?
Results:
If possible, please provide separate information of how many participants answered no or unsure for all tables.
Table.4: Please provide references for the comparison of chest radiographs to CT/NM examinations.
Tables 5-7: please provide percentages.
Discussion:
p.6 L195: results are indeed “alarming” and this should be also mentioned within the conclusion.
p.6 LL184-185: How many examinations on children are performed annually?
p.7 LL231-240: and table 8: please move to results section.
The reviewer would recommend to summarize the authors ideas on how to overcome the problem of lacking awareness on radiation protection within one separate paragraph.
p.8 L254: Could the authors provide any information on how the problem is handled in other countries and if there is any available data on the radiation awareness in those countries?
Conclusions:
In the reviewers opinion there is nothing “quite reassuring” (p8 L259) about the results represented in this study, please consider rephrasing. The authors should draw the conclusion that there is an urgent need for a change in education of medical students and young doctors to improve awareness of radiation exposure on diagnostic radiological examinations.
Author Response
|
No. |
Comments/Suggestions
|
Response |
|
|
Reviewer #1: The provided manuscript by Ng et al. deals with the very interesting topic of awareness of health professionals regarding radiation exposure of diagnostic imaging in pediatric patients. Results of this single-center study are at least unexpected, not to say shocking. In the reviewers opinion it is questionable if the results of this study can be transposed to other hospitals in different countries. However, results are at least very important for the hospital the authors are working in and may could help to raise awareness on radiation protection as well as support the importance of education on radiation protection. Before publishing, the manuscript at hand needs several amendments. Title: good Abstract: p.1 LL30-31 “senior doctors” please provide numbers
p.1 LL32-35 please specify what in your believe are the reasons for the results. Introduction: Please be a little bit more specific on the effects of radiation exposure on children; e.g. “damage at the cellular level”? What effect do you expect at what dose? deterministic and stochastic effects? risk assessment of radiation exposure/ linear no-threshold model, what is the cancer risk in the literature? … Please provide some information on organizational radiation protection measures of your hospital/country; e.g. who can request an examination and who is legally responsible. Methods: Please provide the questionnaire.
To how many health professionals did you send the questionnaire? How many do work within your institution?
Results: If possible, please provide separate information of how many participants answered no or unsure for all tables.
Table.4: Please provide references for the comparison of chest radiographs to CT/NM examinations.
Tables 5-7: please provide percentages. Discussion: p.6 L195: results are indeed “alarming” and this should be also mentioned within the conclusion.
p.7 LL231-240: and table 8: please move to results section. The reviewer would recommend to summarize the authors ideas on how to overcome the problem of lacking awareness on radiation protection within one separate paragraph. p.8 L254: Could the authors provide any information on how the problem is handled in other countries and if there is any available data on the radiation awareness in those countries? Conclusions: In the reviewers opinion there is nothing “quite reassuring” (p8 L259) about the results represented in this study, please consider rephrasing. The authors should draw the conclusion that there is an urgent need for a change in education of medical students and young doctors to improve awareness of radiation exposure on diagnostic radiological examinations.
|
Correction is highlighted in yellow.
The number of seniors doctor has been added to the abstract.
Information has been added to the abstract.
Introduction in the first and second paragraphs has been added as reviewer suggestions. The added information is on page 2.
The information has been added in the last paragraph as per the reviewer's suggestion.
The questionnaire has been added in Appendix 1
The information has been added on page 3. All of them work at our institution.
Most of the participants answer no to the questions. The information has been added to the result.
Information has been added to the result.
The percentage has been added in Tables 5-7
The information has been added in the first paragraph of the discussion.
Table 8 has been removed from the result section.
The separate section has been added as per reviewer suggestions.
Information has been added as per reviewer suggestion in page 8.
A conclusion has been reword as per the reviewer's suggestion. |

Reviewer 2 Report
This paper deals with a survey about awareness of dose exposure in children in health care professionals. The results are depressing an in line with publications done by IAEA -- there is almost no awareness and therefore also no correct patient information.
Legislative bodies should react and appropriate actions in order to ensure that patients and in particular children will get appropriate imaging.
------------------------
Introduction: Well written, the white spot of the scientific map identified.
Material and Methods: A survey represents the backbone of this paper, which consisted of two parts:
A) demographic information of the responder;
B) knowledge based questions about radiation exposure and communication.
Material and methods are clearly understandable. Of course there is no control group and the questions arises how should that group defined.
Results: Detailed statistics are given about both sections of the survey and useful information is extracted. Results have a clear structure and are easy to read. Some of the results could be expected or where already known from other sources, which are referenced.
Discussion: This section is also structured according the purpose of the study.
Conclusions: Good and comprehensive formulation of the main results. I still think, that the paper should be published in order increase the chances that responsible authorities will take appropriate measures and maybe other readers will be motivated to increase their knowledge.
Author Response
|
Reviewer #2:
This paper deals with a survey about awareness of dose exposure in children in health care professionals. The results are depressing an in line with publications done by IAEA -- there is almost no awareness and therefore also no correct patient information.
Legislative bodies should react and appropriate actions in order to ensure that patients and in particular children will get appropriate imaging.
------------------------
Introduction: Well written, the white spot of the scientific map identified.
Material and Methods: A survey represents the backbone of this paper, which consisted of two parts: A) demographic information of the responder; B) knowledge based questions about radiation exposure and communication. Material and methods are clearly understandable. Of course there is no control group and the questions arises how should that group defined.
Results: Detailed statistics are given about both sections of the survey and useful information is extracted. Results have a clear structure and are easy to read. Some of the results could be expected or where already known from other sources, which are referenced.
Discussion: This section is also structured according the purpose of the study.
Conclusions: Good and comprehensive formulation of the main results. I still think, that the paper should be published in order increase the chances that responsible authorities will take appropriate measures and maybe other readers will be motivated to increase their knowledge.
|

Reviewer 3 Report
Kindly see the attached document.

Author Response
|
|
Reviewer #3:
Title: Awareness of Healthcare Professionals on Diagnostic Radiological Examination Related Radiation Exposure in Pediatric Population
Overall comments
This paper is written well but requires some revision to be publishable.
Among the chief areas to consider, the paper is about healthcare professionals yet when one reads it, it only focuses on medical staff (doctors). The use of the term healthcare professionals gives an impression that the study involves different healthcare professionals. The authors need to consider that they are publishing for an international audience and therefore use terminology that is understandable or applicable in various contexts e.g. the authors have also used the term “house officers” and this is not a term used in other contexts – the authors have to find another term or include an explanation of what this terms means.
Medical school curriculum in most countries generally include a radiology rotation / placement. However, on this paper there is no indication of what the curriculum is in Malaysia and whether the doctors that were involved in the study had a radiology rotation during medical school.
Below are the comments specific to the sections of the manuscript.
Title
The title must be revised to be specific with the group of healthcare professionals that were involved in the study i.e. medical doctors. Healthcare professionals generally include other professions like nursing, midwives, radiographers, physiotherapists, dentists, etc. The title will also benefit by including the location of the study. Currently there is no location of the study. The title could therefore be revised to something like “Awareness of Medical Doctors on Diagnostic Radiological Examination Related Radiation Exposure in Pediatric Population in Kuala Lampur.”
NB: The suggestion of the change of the term healthcare professionals must be applied to the whole document.
Abstract
The abstract lack the indication of the location of the study. Further the abstract does not highlight the gap in knowledge that prompted the study to be conducted.
Line 19 – “awareness of healthcare professionals’ awareness…” – this is probably a grammar issue or tautology, the sentence needs to be revised.
Line 21 – “therefore this study…” – this sentence needs to be revised to use correct grammar.
Line 27 – the sentence can be written better to avoid having to say “which is…”
Line 31 – “…this subject matter” – which subject matter is being referred to? This needs to be reworded appropriately.
Lines 31-35 – the whole conclusion sentence needs to be revised for grammar.
Introduction
There are several studies that has been done in various countries regarding the medical doctors ‘ knowledge on radiation protection and related aspects. However, the introduction fails to highlight or refer to any of such studies. This is important because the authors need to be able to demonstrate how their study is contributing new information and how it links with previous similar studies. On the last paragraph the authors have indicated that they included a “quick survey” – what is a quick survey? In the methods section there is nothing indicating a “quick survey” – a quick survey is generally considered a survey that will get immediate results e.g., the participants answering the questions on their devices with the results being displayed on a screen.
My suggestion would be that the authors remove that second part of that paragraph as it does not add value to have it stated.
Line 42 – “diagnostic medical imaging…” the sentence reads as if there is a word missing. I believe it should be diagnostic medical imaging procedures or examinations. And again, in the sentence that follows, these are called radiological procedures – this is lack of consistency. The authors need to decide whether they will call these diagnostic medical imaging procedures or radiological procedures and the decision must be aligned to the context of their study. See also line 44-45.
Lines 51-54 – there is an indication that the radiologists can enforce radiation protection principles. This needs to be expanded upon or clarified because the radiologists are generally not involved in such practice but radiographers or medical imaging technologists are the one who enforce the principles.
Materials and Methods The authors use the word “questionnaires” (see line 75) which gives the impression that there were more than one designed questionnaires. The study setting has not been described adequately. The description of the design of the questionnaire is inadequate. For example, there is no indication of how the questions were formulated i.e. were they based on previous similar studies, literature etc. There is also no indication of whether the questionnaire was piloted in any group, piloting the questionnaire is an important step in its design.
Line 85 – “…in this country…” – which country is being referred to when the location of the study has not been described and the study setting is also not included.
Results
Table 1 shows that 17.9% respondents had their specialization or current posting as “other” – what does other include in the context of the study.
Line 143 – the terms used here are not the terms generally used to describe the types of radiation used. The authors have used the term “dose-saving medical imaging” - the generally acceptable term is non-ionizing. The authors have provided a definition for the dose-saving medical imaging but have not referenced it which might be because this is a definition they created for themselves.
Line 146 – reference is made to 35% on table 2 but I could not find such in the table.
Discussion
Line 216 indicates that the junior doctors are involved in explaining the medical imaging procedure to the parents and patients – is this a correct statement in view of that the procedure is generally explained by the professionals that are doing the actual procedure.
This could be an issue of terminology.
Some results are presented in the discussion section.
The discussion section should be used purely for the purpose of discussing and where relevant reference can be made to results that are already presented in the previous section.
The authors have presented new results under the discussion section – see table 8.
|
Correction is highlighted in blue.
The terms healthcare professional and house officer have been defined in the introduction. Line 82 – 85.
The title has been revised accordingly as per the revierwer suggestion.
The term healthcare professional has been defined in the introduction as per the reviewer's suggestion.
The location has been added as per the reviewer's suggestion.
The line has been corrected.
The line has been corrected.
The line has been corrected.
The line has been corrected.
The line has been corrected.
Previous studies about this radiation protection knowledge are added as per the reviewer's suggestion.
The word quick survey has been deleted as it is confusing.
Corrections and amendments have been done accordingly, as per reviewer suggestions.
Corrections and amendments have been done accordingly, as per reviewer suggestions.
Corrections and amendments have been done accordingly, as per reviewer suggestions.
Corrections and amendments have been done accordingly, as per reviewer suggestions. Missing information has been added accordingly.
Corrections and amendments have been done accordingly, as per reviewer suggestions.
Corrections and amendments have been done accordingly, as per reviewer suggestions
This is usually the practice in our institution.
Results have been taken out from the discussion section as per the reviewer suggestion.
Corrections and amendments have been done accordingly, as per reviewer suggestions
Corrections and amendments have been done accordingly, as per reviewer suggestions Table 8 has been taken out. |

Round 2
Reviewer 1 Report
The amendments made to the manuscript are satisfactory. The manuscript provided by Ng et al. gives interesting insights into radiation awareness regarding diagnostic imaging in a paediatric population within a hospital in Kuala Lumpur. The authors draw a sound conclusion and the presented results could help to raise awareness on radiation protection. The reviewer recommend to accept the manuscript in its present form.
Author Response
Thank you very much for your time and read our manuscript many times.
The manuscript has tremendously improved by your comments and suggestions.
Thank you again

This manuscript is a resubmission of an earlier submission. The following is a list of the peer review reports and author responses from that submission.